# Non-Communicable Disease Burden and Dietary Determinants in Women of Reproductive Age in Sub-Saharan Africa: A Scoping Review

**DOI:** 10.3390/diseases13100313

**Published:** 2025-09-24

**Authors:** Perpetua Modjadji, Ntevhe Thovhogi, Machoene Derrick Sekgala, Kotsedi Daniel Monyeki

**Affiliations:** 1Non-Communicable Diseases Research Unit, South African Medical Research Council, Tygerberg, Cape Town 7505, South Africa; 2Department of Life and Consumer Sciences, College of Agriculture and Environmental Sciences, University of South Africa, Florida Campus, Roodepoort 1709, South Africa; 3Department of Social Sciences, Center for Social Sciences Research (CSSR), Faculty of Humanities, Robert Leslie Social Science Building 12 University Avenue, University of Cape Town, Rondebosch, Cape Town 7701, South Africa; 4Department of Physiology and Environmental Health, School of Molecular and Life Sciences, Faculty of Science and Agriculture, University of Limpopo, Polokwane 0727, South Africa

**Keywords:** non-communicable diseases, dietary determinants, women of reproductive age, sub-Saharan Africa

## Abstract

Background/Objectives: Sub-Saharan Africa (SSA) is experiencing a rising burden of non-communicable diseases (NCDs), projected to surpass infectious diseases as the leading cause of mortality. This shift reflects a complex public health challenge driven by changing dietary patterns and persistent social and gender inequities. Women of reproductive age are particularly vulnerable due to biological and sociocultural factors, with diet playing a central role in NCD development and maternal health. This scoping review explores dietary determinants of NCDs in this population and identifies evidence gaps to support context-specific, gender-responsive interventions. Methods: The review followed the Joanna Briggs Institute methodology and PRISMA-ScR guidelines. A comprehensive search was conducted across PubMed, Scopus, and Google Scholar for studies published between January 2010 and May 2025. After removing duplicates, 577 articles were screened, and 19 met the inclusion criteria. Data were synthesized using descriptive statistics and thematic analysis. An adapted conceptual framework informed by the ecological model was developed to illustrate the multilevel pathways linking dietary determinants to NCD outcomes. Results: Frequent consumption of ultra-processed foods, salty snacks, processed meats, and sugar-sweetened beverages was consistently associated with increased NCD risk. Central obesity was prevalent among nearly half of the women studied, and a high intake of sugary drinks was common across diverse populations. Among pregnant women, overweight was observed in approximately one-quarter of participants, despite the inadequate intake of protein and iron. Vitamin A deficiency was moderately prevalent, and urban residence was linked to a twofold increase in the coexistence of obesity and micronutrient deficiencies. These patterns were shaped by poverty, food insecurity, and the cultural norms influencing dietary behaviors and access to nutritious foods. Conclusion: Dietary determinants significantly contribute to the rising NCD burden among women of reproductive age in SSA, yet adolescent women remain underrepresented in research. Addressing these gaps through culturally sensitive, multisectoral interventions and biomarker-informed longitudinal studies is essential for guiding inclusive policies and sustainable health strategies for this vulnerable population.

## 1. Introduction

Non-communicable diseases (NCDs) are now the leading cause of mortality globally, and sub-Saharan Africa (SSA) faces a dual burden of NCD and infectious diseases [1,2,3,4]. Women of reproductive age (15–49 years) are particularly vulnerable due to intersecting metabolic and reproductive health risks [5,6]. Rising rates of obesity, hypertension, and type 2 diabetes in this group are closely linked to poor dietary quality and metabolic dysfunction, with prevalence varying across SSA countries [7,8,9,10,11]. This growing burden is driven by a regional nutrition transition, from traditional, nutrient-rich diets to energy-dense, ultra-processed foods, often accompanied by reduced physical activity [12,13,14,15,16,17]. Urbanization, food insecurity, and limited access to healthcare and nutrition education further exacerbate these risks [8,9,18,19,20]. Micronutrient deficiencies, especially in vitamin A, iron, and zinc, frequently coexist with overweight and obesity, contributing to adverse maternal and fetal outcomes [21,22,23,24].

Despite increasing research on dietary risk factors for NCDs in SSA, the evidence remains fragmented and methodologically inconsistent, particularly for women of reproductive age, who are central to intergenerational health outcomes. In addition to other dietary determinants, such as high consumption of ultra-processed foods, low dietary diversity, inadequate intake of fruits and vegetables, and reliance on refined carbohydrates, more than two-thirds of women of reproductive age in Africa are estimated to be micronutrient deficient, largely due to widespread poor dietary quality and inadequate intake of nutrient-dense foods to meet their heightened nutritional needs [25].

Dietary determinants are contextual and behavioral factors that influence both food choices and dietary patterns. These encompass individual-level factors such as affordability, nutrition knowledge, and personal preferences; environmental influences including urbanization, food availability, and food insecurity; and sociocultural norms shaped by beliefs, traditions, and gender roles that guide food practices within communities [26,27,28].

Food fortification is a cost-effective and highly recommended food-based approach for addressing these micronutrient deficiencies in low-income settings [25]. Large-scale food fortification programs have shown benefits but are underreported [29,30,31,32]. Studies have recommended targeted fortification strategies for high-risk groups and advancements in fortification technologies, such as microencapsulation to enhance nutrient stability and monitoring systems to improve program coverage and ensure nutrient bioavailability [33,34].

This scoping review applies to the Joanna Briggs Institute methodology [35] and PRISMA-ScR guidelines [36] to map existing evidence and identify gaps. An adapted socio-ecological framework is used to illustrate the multilevel pathways linking dietary determinants to NCD outcomes in this population [16].

## 2. Materials and Methods

### 2.1. Methodological Approach

This scoping review was conducted in accordance with the Joanna Briggs Institute (JBI) methodological guidance for scoping reviews [35] and reported following the PRISMA-ScR (Preferred Reporting Items for Systematic Reviews and Meta-Analyses extension for Scoping Reviews) checklist (Appendix A) [36]. A scoping review was selected to accommodate the broad and exploratory nature of the research question and to map the extent, range, and nature of evidence on dietary determinants of NCDs among women of reproductive age in SSA, where the existing literature is diverse and fragmented. Unlike systematic reviews, which address narrowly defined questions, a scoping review facilitates the mapping of key concepts, evidence types, and research gaps across a complex and heterogeneous field [37,38]. Although the review was not registered with PROSPERO, a preliminary search of the Cochrane Database of Systematic Reviews was conducted to ensure originality and transparency, confirming that no similar reviews had been published. The review process followed the five-stage framework developed by Arksey and O’Malley (2005) [37], with refinements from subsequent methodological advancements:Identifying the research question;Identifying relevant studies;Study selection;Charting the data;Collating, summarizing, and reporting the results.

To ensure conceptual clarity and relevance to the target population and context, the Population–Concept–Context (PCC) framework was used to guide the formulation of eligibility criteria and the scope of the review.

### 2.2. Search Strategy

A comprehensive and systematic literature search strategy (Appendix A) was conducted across three major electronic databases: PubMed, Scopus, and Google Scholar. The search targeted peer-reviewed articles published between January 2010 and April 2025 and was limited to English-language studies with full-text availability. The strategy combined Medical Subject Headings (MeSHs) and free-text terms related to dietary intake, nutritional status, non-communicable diseases, and women of reproductive age in SSA.

To enhance relevance, terms were organized into four thematic categories: population, NCD, dietary determinants, and geographical context. For a study to be considered eligible, it had to include at least one term or phrase from each category. Initially, the search was restricted to titles and abstracts; however, this yielded limited results. The strategy was refined to include full-text searches, which significantly increased the number of eligible studies.

### 2.3. Study Selection

All retrieved citations were imported into EndNote reference management software, where duplicate records were identified and removed. Screening was conducted in two stages:Titles and abstracts were reviewed for relevance;Full-text articles were evaluated against inclusion and exclusion criteria.

Studies were excluded if they focused on infectious diseases, male populations, or were conducted outside SSA. Editorials, reviews, protocols, case reports, and unpublished materials were also excluded.

Two reviewers (N.T. and M.D.S.) independently screened titles, abstracts, and full texts using predefined criteria. Discrepancies were resolved through discussion, and a third reviewer (P.M.) was consulted when necessary to ensure consistency and minimize bias.

#### Eligibility Criteria

Studies were eligible if they fulfilled the following criteria:Focused on women aged 15 to 49 years (women of reproductive age);Examined dietary factors or nutritional status in relation to NCD outcomes;Were conducted in SSA;Used original research designs (cross-sectional, cohort, RCTs, or mixed methods).The MEDLINE search string included combinations of the following terms:Population terms: “Reproductive-Aged Women”, “Women 15–49 Years”, “WRA”, and “female”;NCD terms: “Non-communicable Diseases”, “diabetes”, “hypertension”, “obesity”, and “cardiovascular diseases”;Dietary terms: “Dietary Determinants”, “Nutritional Status”, “Dietary Patterns”, and “Food Security”;Context terms: “Sub-Saharan Africa”, “Southern Africa”, and “South Africa”.

The combined database searches initially yielded 577 articles. After removing duplicates and screening for relevance based on the inclusion and exclusion criteria, 19 studies were identified as eligible for inclusion in the final review. These studies specifically addressed the dietary determinants of NCDs among women of reproductive age in SSA. The characteristics of the included studies were analyzed in terms of their rationale, participant demographics, dietary and health outcomes, study duration, research methodology, and reported impacts. This information was synthesized narratively to maintain a cohesive and contextually grounded presentation of the findings. The study selection process is visually summarized in Figure 1, which presents the PRISMA flow diagram outlining the identification, screening, eligibility, and inclusion stages of the review.

### 2.4. Data Charting and Analysis

Data were extracted using a structured charting form that was developed and iteratively refined by the research team to ensure consistency and relevance. The charting process captured key variables such as study characteristics (including author, year of publication, country, and study design), population demographics, dietary assessment methods, NCD outcomes, and the main findings of each study. Data extraction was conducted independently by two reviewers (N.T. and M.D.S.), with discrepancies resolved through discussion or consultation with a third reviewer (P.M). This approach ensured consistency and minimized bias in the charting process. The analytical approach combined both descriptive numerical summaries and thematic synthesis to provide a comprehensive understanding of the evidence.

Descriptive statistics were employed to quantify the distribution of studies across countries, study designs, and population characteristics, offering a broad overview of the research landscape. In parallel, thematic analysis was conducted to identify recurring patterns and contextual factors that influence dietary behaviors and NCD risk among women of reproductive age. Themes were developed inductively from the data and refined through collaborative discussions among the research team to ensure that they aligned with the objectives of the review. This dual approach enabled the synthesis of both quantitative and qualitative dimensions of the evidence base, yielding valuable insights into the dietary determinants of non-communicable diseases in the target population.

## 3. Results

### 3.1. Study Characteristics

This scoping review synthesized findings from 19 studies conducted across SSA, with South Africa being the most represented country (n = 11), while a small cluster of countries have minimal coverage, including Ghana, Kenya, Burkina Faso, Zambia, Rwanda, Malawi, Nigeria, and Benin. This pattern suggests substantial research inequality across the continent, with evidence concentrated in South Africa, while most of Africa remains understudied. The studies covered a range of geographical contexts, urban, peri-urban, and rural, and diverse methodological designs. These included cross-sectional analyses (n = 15), randomized controlled trials (n = 1), mixed-method approaches (n = 1), and longitudinal cohort studies (n = 2). The study populations were predominantly composed of socioeconomically disadvantaged women, many of whom were unemployed or experiencing food insecurity. For example, in Kenya and Burkina Faso, 38% of participants were classified as severely food insecure. Several studies focused on specific subgroups, such as women living with HIV, women with obesity, pregnant women with micronutrient deficiencies, and urban women at elevated risk of type 2 diabetes. Table 1 reflects the stratification variables, including age group, geographic setting (urban/rural), HIV status, and dietary assessment method, factors that are critical in shaping dietary exposures and NCD risks and that are explicitly discussed in the synthesis. 

### 3.2. Individual-Level Determinants

This section focuses on individual-level dietary determinants, including affordability, nutrition knowledge, and personal food preferences, which directly influence dietary behaviors and NCD risk. A consistent association emerged between the consumption of processed, energy-dense foods and increased risk of NCD. In South Africa, frequent intake of takeaway foods, salty snacks, and processed meats was significantly associated with elevated odds of hypertension, diabetes, and cardiac events (adjusted odds ratios [AORs] ranging from 1.42 to 2.45) [43]. In Rwanda, central obesity (prevalence: 48.5%) was strongly linked to weekly meat consumption (OR = 5.3) and alcohol intake (OR = 5.8) [45].

Sugar-sweetened beverage (SSB) consumption exceeded 50% among women in Kenya and Burkina Faso, with higher intake associated with employment status and greater dietary diversity [44].

Micronutrient deficiencies were also prevalent: inadequate intake of protein, iron, and zinc among pregnant women was associated with adverse fetal outcomes [46], while vitamin A deficiency (11.7% prevalence) was linked to systemic inflammation [47]. Dietary determinants such as food insecurity, cultural norms, and limited nutrition knowledge were frequently reported. In contrast, dietary patterns characterized as Western (high in processed foods and sugary beverages) or traditional (rich in legumes and whole grains) were associated with distinct NCD outcomes. Of the 19 studies included in this scoping review, 11 employed only self-reported dietary assessment methods [39,41,42,44,45,46,47,48,49,50,51], 1 relied exclusively on biomarker-based measures [56], and 4 integrated both dietary questionnaires and biomarker analyses [42,45,47,50]. However, the reliance on self-reported tools indicates a methodological gap as well as the use of cross-sectional studies limiting causal interpretation, as discussed in detail in the limitations section. Two studies did not employ dietary assessment or biomarker-based analysis; instead, they focused on anthropometric outcomes and/or socioeconomic and/or reproductive factors as indirect determinants of nutritional status [8,48]. Additionally, one study used a 24-hour dietary recall method (24HR) to assess dietary quality and NCD-related food patterns, focusing on dietary risk indicators rather than direct clinical NCD outcomes (Table 2).

### 3.3. Socioeconomic and Cultural Influences

This section focuses on the environmental and sociocultural determinants, such as urbanization, food insecurity, and cultural norms, that shape dietary behaviors at the community and societal levels. Socioeconomic deprivation and cultural norms were found to significantly shape dietary behaviors among women of reproductive age in sub-Saharan Africa. In Zambia, women attributed hypertension not only to the consumption of poor-quality foods, such as chemically grown vegetables, but also to psychosocial stress stemming from caregiving responsibilities [41]. In South Africa, cultural perceptions played a notable role, where body fatness was often valorized as a protective social signal against HIV-related stigma. Additionally, unemployment and low income were consistently associated with a reliance on calorie-dense, nutrient-poor diets [8,47]. Urban residence further compounded nutritional risk, with studies showing a twofold increase in the likelihood of coexisting obesity and micronutrient deficiencies among urban women (adjusted prevalence ratio [aPR] = 2.0) [51].

### 3.4. Biological Mediators and Health Outcomes

Nutritional inadequacies during pregnancy emerged as a critical concern across several studies. In South Africa, 24.2% of pregnant women were classified as overweight, yet their intake of essential nutrients such as protein and iron remained below the recommended dietary allowances (RDAs) [46]. Adherence to a traditional dietary pattern, characterized by the consumption of whole grains and legumes, was associated with reduced odds of excessive gestational weight gain (odds ratio [OR] = 0.68) [55]. In contrast, a study evaluating the impact of lipid-based nutrient supplementation (LNS) in Ghana found that such interventions did not significantly reduce the risk of hypertension among pregnant women [40].

### 3.5. Conceptual Framework

To visually synthesize the findings of this scoping review, an adapted conceptual framework was developed to illustrate the pathways linking dietary determinants to NCD outcomes among women of reproductive age in SSA (Figure 2). This framework is informed by the socio-ecological model, which has been previously applied to dietary and physical activity behaviors in similar populations [16]. The framework integrates elements of the ecological model, capturing multilevel influences: individual, interpersonal, community, and societal, and incorporates biomedical mediators such as inflammation and micronutrient status. It also reflects cultural and socioeconomic factors that shape dietary behaviors, including urbanization, poverty, food insecurity, and body image norms. This model provides a structured lens through which to understand the complex interplay of determinants and outcomes, and supports the development of context-specific, culturally responsive interventions.

In this framework, dietary determinants, such as socioeconomic status, food access, and cultural norms, are positioned as upstream influences that shape dietary behaviors. Dietary patterns, including Western and traditional diets, are conceptualized as downstream outcomes of these determinants, which in turn influence NCD risk.

### 3.6. Summary of Evidence

To visually consolidate the thematic findings of this scoping review, a hexagon radial summary figure was developed (Figure 3), linking dietary domains to NCD outcomes. The central hexagon represents NCD outcomes in women of reproductive age, surrounded by six interlinked domains: processed foods, traditional diets, micronutrient deficiencies, socioeconomic factors, cultural norms, and biomarkers. These domains reflect the multilevel and interconnected nature of dietary determinants and their influence on NCD outcomes.

To highlight geographic disparities in research coverage, an African evidence gap map was developed (Figure 4). This visual illustrates the uneven distribution of studies across SSA, with a notable concentration in South Africa (n = 11) and limited representation from other regions (n = 8).

## 4. Discussion

This scoping review synthesizes evidence on dietary determinants contributing to the rising burden of NCDs among women of reproductive age in SSA. The findings showed a complex interplay between poor dietary quality, nutritional inadequacies, and broader socioeconomic and cultural factors. Dietary determinants such as food insecurity, cultural norms, and limited nutrition knowledge influence food choices, while dietary patterns, such as Western (high in processed foods and sugary beverages) or traditional (rich in legumes and whole grains), reflect the habitual combinations of foods consumed. These patterns were associated with distinct NCD outcomes across the studies reviewed. As SSA undergoes a rapid nutrition transition, traditional diets are increasingly replaced by energy-dense, ultra-processed foods, disproportionately affecting women [13,19,24,43,45]. The findings of this review align with a modified ecological framework, which emphasizes the interplay of individual dietary behaviors, interpersonal and cultural norms, community-level food environments, and broader policy influences. This multilevel perspective is essential for designing effective interventions that address both biological and social determinants of NCD risk.

Frequent consumption of processed foods, such as salty snacks, meats, and sugar-sweetened beverages (SSBs), was consistently associated with hypertension, obesity, and type 2 diabetes. These trends mirror global dietary westernization, but are especially concerning in SSA, where health systems are often ill-equipped to manage the dual burden of infectious and chronic diseases. The association between SSB intake and factors like employment and dietary diversity [44] highlights the need for integrated, nutrition-sensitive policies that go beyond economic empowerment to address food environments, marketing, and cultural norms.

A key finding of this review is the widespread coexistence of micronutrient deficiencies with overweight and obesity, a double burden of malnutrition increasingly observed in SSA [8,24,57,58]. Deficiencies in vitamin A, iron, and zinc, particularly among pregnant women, have serious implications for maternal and fetal health [59,60], acting not only as nutritional gaps, but also as biological stressors that exacerbate inflammation and disease risk [61,62]. Despite this, most studies relied on self-reported dietary tools such as 24-hour recalls and food frequency questionnaires, with limited use of biomarker-based assessments, highlighting the need for more objective methods to better understand diet–disease relationships. This shows the need for integrating objective biomarker data in future research to better capture physiological responses to dietary exposures and improve causal inference.

In this context, food fortification emerges as a critical and cost-effective strategy to improve the intake of essential micronutrients among women of reproductive age, particularly in regions where the burden of deficiency is highest [24]. Fortification of staple foods like maize meal and wheat flour with iron, folic acid, and vitamin A has demonstrated positive impacts on reducing deficiencies and improving maternal and child health outcomes [29,32]. However, its potential remains underutilized due to inconsistent implementation, limited public awareness, weak regulatory enforcement, and poor access in rural and informal markets. Furthermore, the lack of disaggregated data on micronutrient status among women of reproductive age limits targeted policy responses. We therefore recommend future studies to conduct stratified analyses to better inform targeted interventions. Stratified analyses by age, geographic location, HIV status, and dietary assessment method are essential to uncover subgroup-specific vulnerabilities and guide the development of tailored nutrition and health policies across diverse SSA contexts. Addressing these barriers is essential to fully leverage food fortification as a scalable and sustainable intervention to reduce micronutrient deficiencies and mitigate NCD risk in this vulnerable population [30].

Dietary behaviors are shaped by socioeconomic and cultural contexts. In Zambia, women linked hypertension to both poor-quality food and caregiving stress [41]. In South Africa, cultural norms that valorize body fatness as protection against HIV stigma continue to influence food choices [63,64]. These findings align with the broader literature on food beliefs and social identity [56,65]. Urbanization, while improving access to services, has also increased exposure to obesogenic environments and processed food markets [24,51]. Nutritional inadequacies during pregnancy, such as low protein and iron intake despite high overweight rates, show the mismatch between dietary patterns and physiological needs [46]. Conversely, traditional diets rich in legumes and whole grains were protective against excessive gestational weight gain [55]. In addition to individual and community-level factors, structural interventions targeting the broader food environment are critical. Evidence from SSA and other LMICs suggests that policies such as taxation on sugar-sweetened beverages, front-of-pack labeling, and restrictions on marketing unhealthy foods to children can significantly influence dietary choices and reduce NCD risk [18,66].

The included studies offer several strengths that contribute to the overall value of this review. Many studies employed validated dietary assessment tools and included diverse populations across urban and rural settings. A subset incorporated biomarker-based measures, enhancing the objectivity of nutritional status assessments. Additionally, the geographic spread, though uneven, provides insight into regional dietary patterns and NCD risks among women of reproductive age in SSA.

Further identified key methodological and contextual limitations highlight the need for standardized data collection in future cohort studies. To address these gaps, future studies should consider collecting a standardized minimum dataset that includes dietary diversity scores (DDS), biomarkers such as C-reactive protein (CRP) and HbA1c, and anthropometric indicators to improve data quality and enable cross-study comparisons. To support systematic and comparable research across sub-Saharan Africa (SSA), we propose a minimum data package for NCD-focused nutrition studies. This should include the following: (1) standardized biomarkers such as CRP, HbA1c, and iron indices to assess inflammation and metabolic risk; (2) anthropometric indicators including BMI, waist circumference, and body composition measures; (3) DDS and context-adapted dietary assessment tools; and (4) sociodemographic variables such as age, education, income, and urban/rural residence. Collecting these core data elements will improve comparability, enable stratified analyses, and strengthen the evidence base for targeted interventions. The predominance of South African studies, driven by stronger research infrastructure and national datasets, raises concerns about regional equity in evidence generation, making it difficult to generalize the findings to other African regions. Geographic data gaps may lead to an overreliance on evidence from a limited number of countries or urban settings, potentially resulting in policy blind spots and interventions that fail to address the diverse dietary patterns and health needs of underserved populations. Limited representation from other SSA countries creates gaps in understanding dietary determinants across diverse sociocultural contexts, potentially skewing policy relevance. Future research should prioritize underrepresented regions and populations to ensure inclusive, context-specific interventions. Many studies had small sample sizes, short durations, and relied on cross-sectional designs and self-reported dietary data, which limit causal inference and introduce bias. Future research should also prioritize validating and adapting dietary assessment tools for use in low-literacy and resource-limited settings to ensure accurate data collection and culturally appropriate nutrition interventions. To improve rigor and applicability, longitudinal designs and biomarker-based assessments should be adopted. The quality of evidence, particularly the predominance of cross-sectional designs and reliance on self-reported dietary data, has direct implications for implementation feasibility. These limitations constrain the precision of risk estimates and the ability to design targeted interventions. Strengthening evidence through longitudinal studies and biomarker-informed assessments will enhance the reliability of findings and support the development of scalable, context-specific nutrition policies. To improve rigor and applicability, future cohort studies should incorporate standardized biomarkers, such as CRP, HbA1c, and iron indices, to assess inflammation and metabolic risk. Additionally, dietary assessment tools should be culturally and contextually adapted to ensure accurate data collection across diverse SSA populations. Given the unique biological and sociocultural vulnerabilities of women of reproductive age, interventions must be gender-responsive, integrating reproductive health with nutrition and NCD prevention.

Multisectoral collaboration across health, agriculture, education, and urban planning is essential to improve food environments and promote equitable access to nutritious diets. These recommendations align existing policy instruments such as the WHO Package of Essential Noncommunicable Disease Interventions (WHO PEN) with national dietary guidelines, which advocate improved dietary quality, reduced intake of ultra-processed foods, and the integration of nutrition services into primary healthcare. Nutrition policies should focus on reducing ultra-processed food intake and integrating dietary counseling into maternal care. We recommend leveraging the role of community health workers to deliver culturally appropriate nutrition education and counseling, particularly in underserved and resource-limited settings. Urban-specific strategies and biomarker-informed research are vital to address obesity, micronutrient deficiencies, and identify at-risk subgroups. Adolescent women of reproductive age were notably underrepresented in the included studies, and where they were included, age-disaggregated data were often lacking. This limits the ability to assess adolescent-specific dietary patterns, nutritional risks, and NCD vulnerabilities, despite evidence that adolescence is a critical window for establishing lifelong health behaviors and addressing intergenerational health risks. Future research should prioritize adolescent-focused analyses and interventions, incorporating age-specific dietary assessments and biomarker data to inform targeted strategies that address the unique biological, social, and developmental needs of this subgroup.

Additionally, national and regional nutrition strategies should prioritize the scale-up and monitoring of food fortification programs, particularly in underserved areas, to address persistent micronutrient deficiencies and reduce NCD risk among women of reproductive age. Furthermore, the emerging use of digital tools and mobile health platforms to support dietary behavior changes and improve access to nutrition services, especially among younger and urban populations, should be explored.

## 5. Conclusions

This review highlights the pivotal role of diet in the rising burden of NCDs among women of reproductive age in SSA. Addressing both the quantity and quality of dietary intake is essential to improve individual health outcomes and mitigate intergenerational risks. The evidence highlights the urgent need for context-specific, culturally appropriate, and multisectoral interventions that integrate health, agriculture, education, and urban planning to cultivate supportive food environments and promote sustainable dietary behaviors. Future research should adopt longitudinal cohort designs for causal effects and incorporate biomarker-based assessments to better capture the complex relationships between diet, metabolic health, and disease risk. Specifically, we recommend the inclusion of standardized biomarkers such as CRP, HbA1c, and micronutrient panels to strengthen the evidence base and inform more targeted, context-specific interventions. These approaches will generate actionable insights to inform policies tailored to the diverse needs of women across SSA. Efforts are especially critical in underrepresented regions, where gaps in research and policy implementation persist. Integrating evidence-based dietary strategies, including the scale-up of food fortification programs, into broader health and development agendas can help to reduce NCD prevalence and support long-term well-being among women of reproductive age.

## Figures and Tables

**Figure 1 diseases-13-00313-f001:**
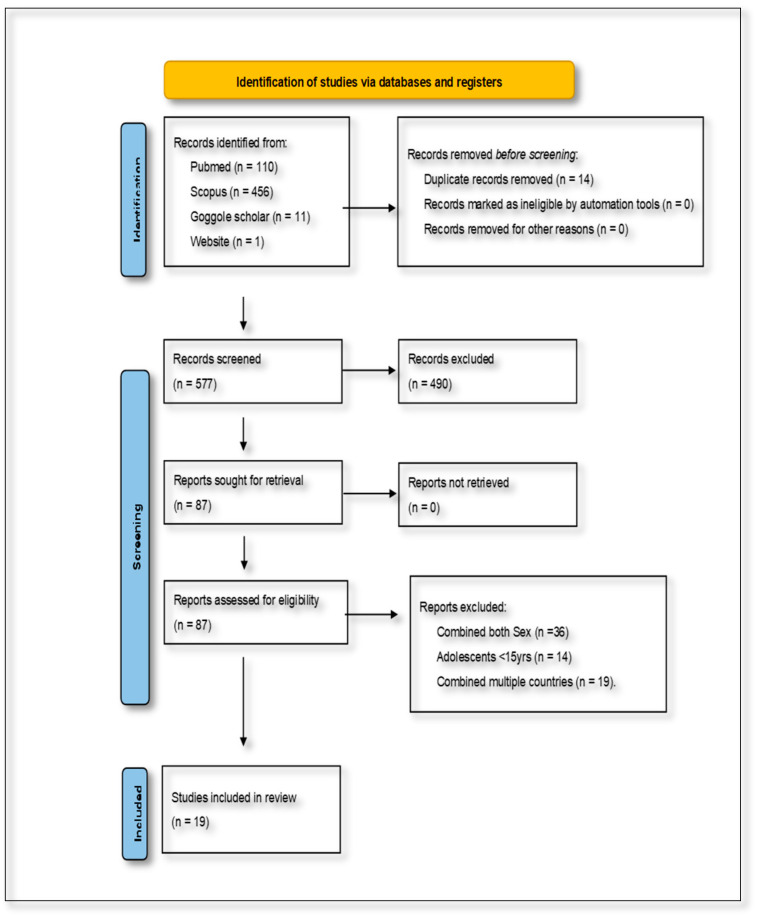
Preferred Reporting Items for Systematic Reviews and Meta-Analyses flow diagram, showing the selection of studies for the present scoping review.

**Figure 2 diseases-13-00313-f002:**
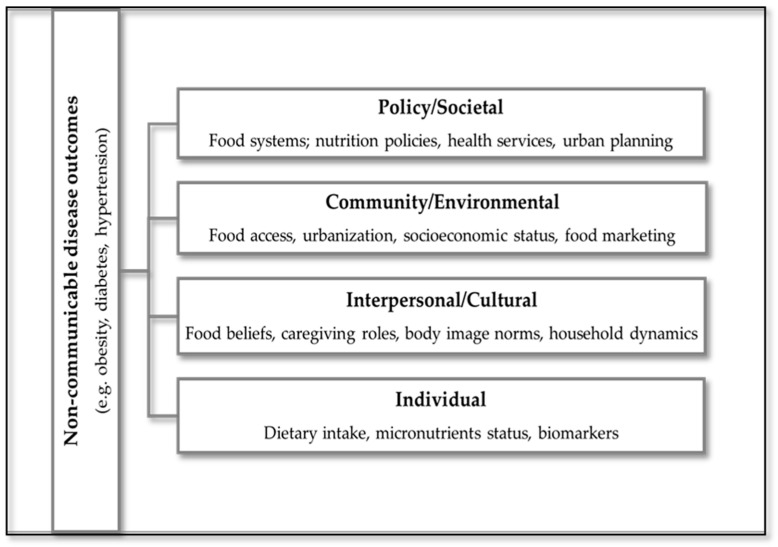
Conceptual framework illustrating multilevel dietary determinants and their pathways to NCD outcomes among women of reproductive age in SSA.

**Figure 3 diseases-13-00313-f003:**
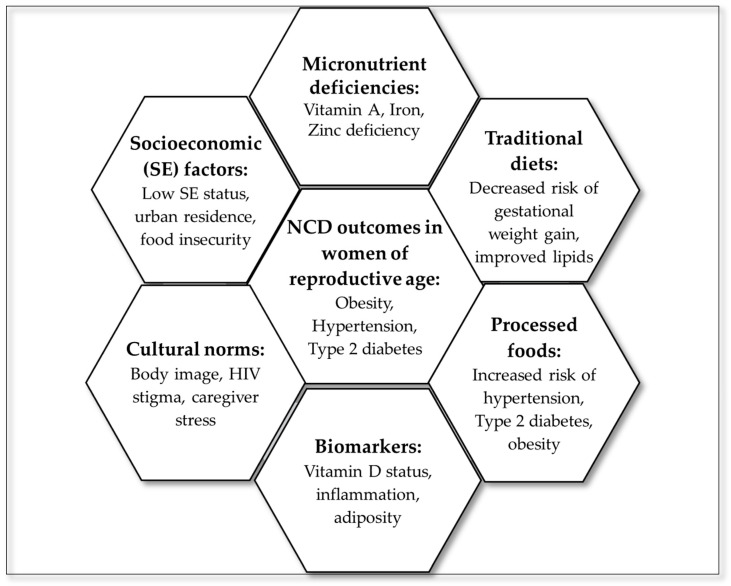
Summary of evidence from the scoping review.

**Figure 4 diseases-13-00313-f004:**
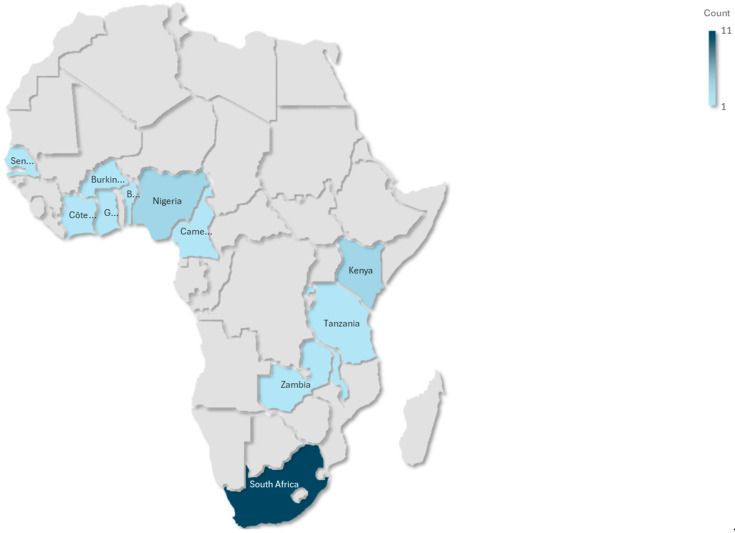
African evidence map.

**Table 1 diseases-13-00313-t001:** The characteristics of the included studies.

Author(s), Year	Country/Region	Life Stage	Study Setting	Study Design	HIV Status	Population Characteristics
Hoosen et al., 2024 [39]	South Africa (Khayelitsha, Cape Town)	Non-pregnant/non-lactating women (20–45 years)	Community health center in a resource-limited urban settlement (peri-urban)	Mixed-method, feasibility pilot trial with in-depth interviews (IDIs) and focus group discussions (FGDs); 4-week single-arm intervention	Living with HIV	33 isiXhosa-speaking women, BMI ≥ 25 kg/m^2^, on dolutegravir-based ART, mostly unemployed, living in informal housing, with moderate food insecurity
Abreu et al., 2021 [40]	Ghana, West Africa	Pregnant women (≥18 years)	Semiurban prenatal clinics in Yilo and Manya Krobo districts	Randomized controlled trial and prospective cohort analysis	HIV negative	1320 pregnant women (≤20 weeks gestation, no severe illness, living ≤20 km from clinic)
Tateyama et al., 2019 [41]	Zambia (Mumbwa District, Central Province)	Non-pregnant/non-lactating women (aged 40+)	Rural community setting, Mumbwa Township Clinic catchment area	Qualitative study (in-depth interviews and focus group discussions)	Living with HIV	67 adults (40 women); mostly Christian, many widowed, with low education and income levels. 32.8% reported hypertension, 6% diabetes, 9% history of stroke.
Oldewage-Theron & Egal A, 2013 [42]	South Africa (Qwa-Qwa, Free State Province)	Non-pregnant/non-lactating women (aged 19–75 years)	Peri-urban community setting	Single-system longitudinal design over 18 months	Not specified	90 Sotho-speaking women, low-income (household income < ZAR 2000), not on lipid-lowering medication, with 40% classified as hypercholesterolemic
Godbharle et al., 2024 [43]	South Africa (nationally representative sample)	Adolescents (aged 15+)	Household-based; urban and rural areas across all nine provinces	Cross-sectional analysis of secondary data from SADHS 2016	HIV negative	10,336 adults, of which 59.3% (n ≈ 6126) were women; mostly Black African, diverse in education and wealth index
Semagn et al., 2023 [44]	Kenya and Burkina Faso (sub-Saharan Africa)	Adolescents and adults (aged 15–49 years)	Household-based survey across rural and urban areas	Cross-sectional study using secondary data (IPUMS-PMA 2018)	HIV negative	3759 women; median age 27; 72.72% unemployed; 38% severely food-insecure households
Kantarama et al., 2023 [45]	Rwanda	Postpartum women (aged 18–45 years)	Two public family planning centers in Kigali City	Cross-sectional study	HIV negative	138 women, non-pregnant, physically healthy, not on chronic disease treatment; 71% ate meat weekly, 75% sedentary lifestyle, 66.7% breastfeeding, 71% not consuming alcohol
Motadi et al., 2020 [46]	Vhembe District, Limpopo Province, South Africa	Pregnant (ages 15–50 years)	Rural area; 16 clinics in Vhembe District	Cross-sectional descriptive study	HIV negative	240 pregnant women; 78% had secondary education, 77% unemployed; 52.5% received iron, folate, and calcium supplements
Parker et al., 2017 [47]	South Africa	Non-pregnant/non-lactating women (aged 16–35 years)	National household survey	Cross-sectional survey (secondary analysis of SANHANES-1 data)	Not specified	Non-pregnant women (n = 1205); 85% Black African, 53.5% urban formal residents, 35% low-income households
Modjadji, 2020 [8]	South Africa (Limpopo Province)	Postpartum women (mean age 37 ± 7 years)	Rural Dikgale Health and Demographic Surveillance System (HDSS) Site	Cross-sectional study	Not specified	508 mothers of primary school children; 63% single, 82% unemployed, 41% low literacy
Said-Mohamed et al., 2018 [48]	South Africa (Urban Soweto, Johannesburg, rural Agincourt HDSS, Mpumalanga)	Adolescents (aged 21–23 and aged 18–21)	Urban (Soweto) and rural (Agincourt HDSS) communities	Cross-sectional study	Not specified	Sample Size: 482 urban women and 509 rural women Demographics: Black South African women; urban participants from the Birth-to-Twenty cohort, rural participants from Agincourt HDSS Key Traits: Age at menarche, adult height, leg-length, waist circumference
Alaofè & Asaolu, 2019 [49]	Benin (Kalalé District, northern region)	Postpartum women (aged 15–49)	Rural households across 16 villages (intervention and control)	Cross-sectional study	HIV negative	426 non-pregnant women with children aged 6–59 months; 15.5% overweight/obese, 8.2% underweight, mostly involved in agriculture or business occupations
Mtintsilana et al., 2019 [50]	South Africa (Soweto, Johannesburg)	Non- pregnant/non-lactating women (median age 53)	Urban community-based (Birth-to-Twenty Plus cohort)	Cross-sectional study	HIV negative	190 middle-aged Black South African women, non-pregnant, not diabetic, residing in Soweto
Rhodes et al., 2020 [51]	Malawi	Adolescents (aged 15–49 years)	National survey (urban and rural areas)	Cross-sectional analysis of the 2015–2016 Malawi Micronutrient Survey (MNS)	Not specified	Non-pregnant women (n = 723); 90.9% rural, 9.1% urban; 79% with less than secondary education
Prioreschi et al., 2021 [52]	Soweto, South Africa	Adolescents (aged 18–25 years)	Urban area with formal and informal housing; high population density (6357 people/km^2^)	Cross-sectional household survey	Not specified	Women (n = 1698); 44% overweight/obese; majority had completed high school; 38% unemployed; low-leisure-time physical activity
Soepnel et al., 2023 [53]	South Africa (Soweto)	Adolescents (aged 18–25 years)	Urban, low-income setting	Cross-sectional sub-study of the Healthy Life Trajectories Initiative (HeLTI) pilot trial	Living with HIV	493 women; median age 21 years; 27.6% vitamin D deficient, 5.6% deficient; 39.1% anemic; 37.5% iron deficient
Mosuro et al., 2023 [54]	Ibadan, Oyo State, Nigeria	Non-pregnant/non-lactating women (aged 45–60 years)	Urban (Felele) and peri-urban (Oje) communities in Ibadan	Cross-sectional survey	Not specified	300 female adults (150 from each location)
Wrottesley et al., 2017 [55]	South Africa (Soweto, Johannesburg)	Pregnant women (aged ≥18 years)	Chris Hani Baragwanath Hospital, Soweto—urban, poor African context	Longitudinal cohort study (Soweto First 1000-Day Study)	Living with HIV	538 pregnant urban Black South African women; <20 weeks gestation; singleton, naturally conceived pregnancies
Janmohamed et al., 2024 [29]	Cameroon, Côte d’Ivoire, Kenya, Nigeria (Adamawa, Benue, Nasarawa states), Senegal, Tanzania	Women of reproductive age (15–49 years)	National and sub-national surveys in rural and urban areas	Cross-sectional, multi-country dietary survey	Not specified	N = 16,584 women, with varied education, rural/urban residence, and household wealth levels

**Table 2 diseases-13-00313-t002:** Mapping of dietary factors, assessment methods, and associated NCD outcomes across included studies.

Author(s), Year	Dietary Focus	Dietary Assessment Method	NCD Outcomes Investigated	Key Findings	Risk/Association Measures
Hoosen et al., 2024 [39]	Time-restricted eating (TRE)	Qualitative self-report via interviews and use of a daily eating window calendar.	Weight perception, appetite control, energy levels, eating habits; no direct clinical NCD biomarkers measured in this pilot.	TRE improved energy levels, weight control, and appetite regulation. Cultural barriers noted.	Qualitative benefits; 100% retention in trial.
Abreu et al., 2021 [40]	Nutrient supplements (LNS)	Intervention-controlled supplementation with biweekly home visits for adherence reporting and unconsumed supplement counts.	Maternal hypertension (systolic BP ≥ 130 mm Hg or diastolic BP ≥ 80 mm Hg).	LNS did not reduce maternal BP; high DBP linked to low birth weight and preterm birth.	RR for LBW = 2.58; RR for PTB = 3.30.
Tateyama et al., 2019 [41]	High salt/sugar/oil intake	Qualitative self-report during interviews and field observations.	Perceived risk and experience of hypertension, stroke, diabetes, obesity; body image beliefs related to HIV stigma; no biomarker measurements conducted.	Diet, stress, and poverty linked to hypertension and stroke. Cultural norms influenced dietary choices.	Qualitative links between stress, diet, and NCD risk.
Oldewage-Theron & Egal A, 2013 [42]	Soybean consumption	3-day 24-hour dietary recall interviews (multi-pass), analyzed with FoodFinder^®^ software, version 3.0.	Blood lipids (total cholesterol, HDL-C, LDL-C, triglycerides), BMI (obesity prevalence).	Improved LDL-C and HDL–LDL ratio in hypercholesterolemic women. HDL-C remained low.	LDL-C decreased (5.4 to 3.9 mmol/L, *p* = 0.032).
Godbharle et al., 2024 [43]	Processed food intake	Structured survey questionnaire; frequency-based food consumption self-reported.	Self-reported diagnosis of hypertension, cardiac arrest, stroke, cancer, hypercholesterolaemia, diabetes, asthma, chronic bronchitis.	Processed foods increased the odds of hypertension, diabetes, and cardiac arrest.	AOR: takeaway foods and hypertension = 1.42 (*p* < 0.05).
Semagn et al., 2023 [44]	Sugar-sweetened beverages (SSBs)	24-hour dietary recall on SSBs and snack consumption; MDD assessed via 10-group dietary diversity score.	Not directly measured; study investigated risk behaviors linked to NCD (SSB intake as proxy for diet-related NCD risk)	Higher SSB intake is linked to education, employment, and snack consumption.	AOR: Primary school = 1.35; Secondary = 1.46.
Kantarama et al., 2023 [45]	Meat, alcohol, coffee intake	Structured interviews with frequency-based recall (past 4 weeks).	Central obesity, lipid profile (TC, HDL, LDL, TG), blood pressure (SBP, DBP), HbA1C, hs-CRP.	Central obesity associated with age, alcohol, and meat intake (OR = 5.3).	OR for meat intake = 5.3 (*p* < 0.05).
Motadi et al., 2020 [46]	Micronutrient deficiencies	Food frequency questionnaire (FFQ), adapted from the National Food Consumption Survey (2005) to include indigenous foods.	Maternal malnutrition (underweight, overweight, obesity), micronutrient deficiencies, pregnancy complications (e.g., preterm birth, low birth weight).	Low protein/micronutrient intake linked to poor fetal development.	Unemployment and low SES correlated with malnutrition.
Parker et al., 2017 [47]	Vitamin A deficiency	Qualitative food frequency questionnaire (FFQ) with 7-day recall.	Vitamin A deficiency, inflammation status (CRP levels).	Vitamin A deficiency prevalence higher in Black women and low-income households.	OR for Black race = 1.89 (*p* = 0.031).
Modjadji, 2020 [8]	Household determinants	Not specified.	Overweight, obesity, abdominal obesity (WC, WHR, WHtR).	Spouse-headed households and multiple pregnancies linked to obesity.	OR for spouse-headed households = 3.5 (95% CI: 1.97–6.31).
Said-Mohamed et al., 2018 [48]	Urban vs. rural diets	Not specified.	Abdominal adiposity (waist circumference).	Urban women had earlier menarche and shorter stature, but similar waist circumference.	β = −2.41 for urban waist circumference (95% CI: −3.31 to −1.51).
Alaofè & Asaolu, 2019 [49]	Dietary diversity	24-hour dietary recall; dietary diversity score (DDS).	Overweight/obesity in mothers (BMI ≥ 25), underweight (BMI < 18.5); co-occurrence with child undernutrition as a form of household double burden of malnutrition (DBM).	Higher SES and TV exposure linked to obesity; education protective.	AOR for SES = 4.82; TV watching = 3.19.
Mtintsilana et al., 2019 [50]	Pro-inflammatory diet	7-day food frequency questionnaire (FFQ), analyzed via validated DII scoring algorithm.	Markers of type 2 diabetes (T2D) risk: fasting glucose, insulin, HbA1c, 2-hour OGTT glucose, insulin sensitivity (Matsuda Index), HOMA2-IR; inflammatory cytokines: TNF-α, IL-8, MCP-1.	Higher E-DII linked to T2D risk markers; visceral adipose tissue mediated effects.	β = 1.70 for fasting insulin (*p* = 0.008).
Rhodes et al., 2020 [51]	Micronutrient deficiencies	Biomarker analysis (serum samples for micronutrients, hemoglobin for anemia), anthropometry (BMI); no direct dietary intake data collected.	Overweight/obesity (BMI ≥ 25 kg/m^2^) and co-occurring micronutrient deficiencies or anemia (double burden of malnutrition, DBM).	Urban women had higher prevalence of obesity and micronutrient deficiency.	aPR for urban residence = 2.0 (95% CI: 1.1–3.5).
Prioreschi et al., 2021 [52]	Food insecurity	Questionnaire (adapted CCHIP index for food insecurity); self-reported street vendor use frequency; household dietary responsibility scores.	Obesity (BMI ≥ 30 kg/m^2^), overweight (BMI 25–29.9 kg/m^2^).	Street vendor use reduced obesity; SES are linked to food insecurity.	B = −0.236 for street vendor use (*p* < 0.05).
Soepnel et al., 2023 [53]	Vitamin D deficiency	Questionnaire (three questions on household food insecurity).	Vitamin D deficiency, iron deficiency, anemia, adiposity (Fat Mass Index, FMI).	No link between vitamin D and anemia; FMI inversely related to vitamin D.	B = −0.01 for FMI (95% CI: −0.016 to −0.003).
Mosuro et al., 2023 [54]	High meat/starchy foods	Food frequency questionnaire (FFQ) and interviewer-administered questionnaire.	Overweight and obesity (BMI, waist–hip ratio).	Overweight (45.4%) and obesity (31.6%) are linked to meat/poultry intake.	Significant association with age (*p* < 0.05).
Wrottesley et al., 2017 [55]	Dietary patterns (Western, traditional)	Interviewer-administered quantitative food frequency questionnaire (QFFQ).	Gestational weight gain (GWG), BMI-specific weight gain categories (inadequate, adequate, excessive) GWG is a predictor of future risk of obesity, T2DM, and CVD.	Western diet increased GWG; traditional diet reduced excessive GWG.	OR for Western diet = 1.30 (*p* = 0.014).
Janmohamed et al., 2024 [29]	Dietary quality and NCD-related food patterns	24-hour recall (DQQ tool).	No specific clinical NCD outcomes measured; instead, dietary risk indicators (e.g., NCD-risk score, NCD-protect score) were assessed as proxies for NCD risk.	Low dietary diversity in Tanzania (43%) and Cameroon (46%); high sweet consumption (up to 80%); low vitamin A-rich food and egg intake.	Higher MDD-W associated with urban residence and secondary+ education (Kenya OR = 10.40); rural residence linked to lower MDD-W.

## Data Availability

The original contributions presented in the study are included in the article/Appendix A; further inquiries can be directed to the corresponding author.

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
