# Peer review of "Non-Communicable Disease Burden and Dietary Determinants in Women of Reproductive Age in Sub-Saharan Africa: A Scoping Review"

_diseases, 2025, doi:10.3390/diseases13100313_

Round 1
Reviewer 1 Report
Comments and Suggestions for Authors
Interesting Scoping Review with dietary factors associated with chronic diseases, methodologically I find it adequate, however, the summary tables are the main weakness of this manuscript, they are difficult to read, they contain a lot of text. My suggestion is to simplify them, to leave the essential information, to arrange the content of the tables according to topic or by order of references.
I also suggest adding a summary figure of the evidence found in this review.
Author Response
Reviewer Comment: Interesting Scoping Review with dietary factors associated with chronic diseases, methodologically I find it adequate, however, the summary tables are the main weakness of this manuscript, they are difficult to read, they contain a lot of text. My suggestion is to simplify them, to leave the essential information, to arrange the content of the tables according to topic or by order of references.
Response: We appreciate your feedback on the summary tables. We have simplified the tables to enhance readability by focusing on essential information and arranging the content according to topic and order of references. Table 1 (Lines 287-289) and Table 2 (lines 298-299)
Reviewer Comment: I also suggest adding a summary figure of the evidence found in this review.
Response: Thank you for the suggestion. We have added a summary figure (Figure 3, lines 269-278) to visually consolidate the evidence found in this review.
Reviewer 2 Report
Comments and Suggestions for Authors
Review: Diseases-3729774
Title: Non-Communicable Disease Burden and Dietary Determinants in Women of Reproductive Age in Sub-Saharan Africa: A Scoping Review
Dear Editors and Author:
This is a timely and well-structured scoping review that addresses a critical public health issue in sub-Saharan Africa. The authors adopted appropriate methodological guidelines and provided a comprehensive synthesis. However, small revisions are suggested to improve analytical depth, especially in the discussion, and to increase methodological transparency.
My Suggestions:
1-Add clarity about the number of reviewers and the data extraction process.
2.Reduce redundancy in the introduction.
3.Discuss more critically the predominance of studies from South Africa.
4.Strengthen the conclusion by addressing the urgency of multisectoral strategies.
Author Response
Reviewer Comment: This is a timely and well-structured scoping review that addresses a critical public health issue in sub-Saharan Africa. The authors adopted appropriate methodological guidelines and provided a comprehensive synthesis. However, small revisions are suggested to improve analytical depth, especially in the discussion, and to increase methodological transparency.
Comment: The English could be improved to more clearly express the research.
Response: The manuscript has been proofread.
My Suggestions:
-
Add clarity about the number of reviewers and the data extraction process.
- Response: We appreciate this suggestion and have revised the Data Charting and Analysis section to clarify the data extraction process. Specifically, we have added that data extraction was conducted independently by two reviewers (NT and MDS), with discrepancies resolved through discussion or consultation with a third reviewer (PM) (lines 128-131).
-
Reduce redundancy in the introduction.
- Response: We have revised the Introduction section to reduce redundancy and improve flow (lines 44-76). Repetitive statements regarding the dual burden of disease and dietary transitions have been consolidated. The revised version maintains all key references while presenting the background more concisely and clearly.
-
Discuss more critically the predominance of studies from South Africa.
- Response: We acknowledged the predominance of studies from South Africa and have addressed this more critically in the Discussion section. We noted that while South Africa’s stronger research infrastructure contributes valuable insights, this geographic concentration may limit the generalizability of findings across SSA. We emphasize the need for future research to include underrepresented regions to better reflect the diversity of sociocultural and dietary contexts in SSA (lines 362-372).
-
Strengthen the conclusion by addressing the urgency of multisectoral strategies.
- Response: We agree with the importance of multisectoral approaches and have revised the Conclusion to reflect this. The updated conclusion emphasizes the need for integrated strategies involving health, agriculture, education, and urban planning to address dietary determinants of NCD. These coordinated efforts are essential to improve health outcomes and reduce intergenerational risks among women in SSA (lines 399-413).
Reviewer 3 Report
Comments and Suggestions for Authors
This manuscript offers a valuable synthesis of dietary risk factors and NCD outcomes among women of reproductive age in sub-Saharan Africa, showing commendable alignment with PRISMA-ScR and JBI standards. Its contextual sensitivity to socioeconomic and cultural determinants is a strength. However, to enhance its scientific and translational value, several refinements are needed:
- The dataset is heavily skewed toward South Africa (10 of 18 studies), while key West and Central African countries (e.g., Cameroon, Senegal, Nigeria) are missing.
- Adolescent WRA are insufficiently examined, despite distinct vulnerabilities.
- Important syntheses are omitted, such as Janmohamed et al. (2024), which overlaps thematically.
- No conceptual model is provided to link determinants and outcomes (e.g., ecological or PEN-3 frameworks).
- Intervention recommendations remain vague, lacking actionable guidance.
- Key regional programs (food fortification, CHW-led nutrition, digital tools) are not considered.
- The appropriateness of dietary assessment tools in low-literacy settings is not evaluated.
Recommendations for strengthening the manuscript:
- Include the full search string and exclusion rationale in an appendix.
- Add a simple risk-of-bias matrix or heatmap.
- Distinguish dietary determinants (e.g., access, knowledge) from dietary patterns (e.g., Western, traditional).
- Stratify findings by life stage (e.g., adolescence, pregnancy, postpartum).
- Use biomarker data to validate self-reported dietary intake.
- Recommend a minimum dataset for future cohort studies (e.g., DDS, CRP, HbA1c).
- Address how geographic data gaps affect equity in research and policymaking.
- Propose a visual evidence gap map (e.g., traffic light system by country/theme).
Methodological gaps relative to JBI 2020 guidance:
- No publicly available protocol (e.g., OSF registration or retrospectively to PROSPERO).
- No justification for selecting a scoping review over a systematic review.
- Lacks documentation of pilot testing for inclusion criteria or data extraction.
- Does not report excluded studies with reasons at full-text screening.
- No evidence of handsearching or updating the search iteratively.
- Unclear whether data extraction involved multiple reviewers or how disagreements were resolved.
- No structured synthesis framework (e.g., mapping by PCC or life stage).
- Exclusion of non-English studies is unjustified.
- Lacks visual/tabular synthesis of core concepts and contexts.
- No critical appraisal, even informally, to guide interpretation.
Addressing these areas would significantly enhance the review’s methodological fidelity, representativeness, and policy relevance.
Author Response
Comment 1: The dataset is heavily skewed toward South Africa (10 of 18 studies), while key West and Central African countries (e.g., Cameroon, Senegal, Nigeria) are missing.
Response: We appreciate the reviewer’s observation regarding the geographic distribution of included studies. We have revised the manuscript to explicitly acknowledge this limitation in the Discussion section (lines 362-372). Specifically, we now state that the dataset is heavily skewed toward South Africa (11 of 19 studies), with limited representation from key West and Central African countries such as Cameroon, Senegal, and Nigeria. We further emphasize that this imbalance may limit the generalizability of our findings to other regions within sub-Saharan Africa. This limitation shows the need for more regionally diverse research to inform inclusive and context-specific interventions across the continent.
Comment 2: Adolescent WRA are insufficiently examined, despite distinct vulnerabilities.
Response: We fully acknowledge the importance of disaggregating data to better understand the unique nutritional vulnerabilities of adolescent women of reproductive age. While we aimed to stratify our findings by age group where possible, this was not feasible due to limitations in the included studies. Most studies in our review focused on women aged 18–49 years, with only two studies including participants as young as 15 years. However, these studies did not provide age-disaggregated data that would allow for meaningful stratification. We recognize the value of such an approach and strongly recommend that future research prioritize adolescent-specific analyses. Hence, we have added that “In particular, adolescent women of reproductive age (15–19 years) were underrepresented in the included studies, and where they were included, age-disaggregated data were not reported. This limited our ability to analyze adolescent-specific dietary patterns and NCD risks. We recommend that future research prioritize adolescent-focused analyses to better understand and address the unique vulnerabilities of this critical life stage.” Lines 388-393
Comment 3: Important syntheses are omitted, such as Janmohamed et al. (2024), which overlaps thematically.
Response: Janmohamed et al. (2024), is now included and the data has been extracted.
Comment 4: No conceptual model is provided to link determinants and outcomes (e.g., ecological or PEN-3 frameworks).
Response: We have developed and included Figure 2, an adapted conceptual framework informed by the socio-ecological model. This framework illustrates the multilevel pathways linking dietary determinants to NCD outcomes among women of reproductive age in SSA. It integrates individual-level dietary behaviors, interpersonal and cultural norms, community-level food environments, and broader policy influences, along with biomedical mediators such as inflammation and micronutrient status. This addition strengthens the theoretical grounding of the review and supports the interpretation of findings within a structured, context-specific lens.
Comment 5: Intervention recommendations remain vague, lacking actionable guidance.
Response: Actionable recommendations are modified and clear (Lines 359 - 398)
Comment 6: Key regional programs (food fortification, CHW-led nutrition, digital tools) are not considered.
Response: We have expanded the discussion section to explicitly acknowledge and integrate key regional strategies that are relevant to addressing dietary determinants of NCD among women of reproductive age in SSA. Specifically, we now highlight:
- Food fortification as a cost-effective and scalable intervention, with examples of its implementation in South Africa and other SSA countries (see lines 69-72) and its impact in lines 334-343.
- The role of community health workers in delivering culturally appropriate nutrition, education and counseling, particularly in underserved areas (see lines 384-386).
- The emerging use of digital tools and mobile health platforms to support dietary behavior change and improve access to nutrition services, especially among younger and urban populations (see lines 396 -398).
Comment 7. The appropriateness of dietary assessment tools in low-literacy settings is not evaluated.
Response: We acknowledge the importance of evaluating the appropriateness of dietary assessment tools, especially in low-literacy settings where tool usability and comprehension are critical. While Table 2 provides a detailed summary of the dietary assessment methods used across the included studies, the primary aim of our scoping review was to map the evidence on dietary determinants of NCDs rather than to critically appraise or evaluate the validity or suitability of the tools themselves. Nonetheless, we agree that tool appropriateness is a crucial aspect in such contexts, and we now highlight this as an important evidence gap in the discussion section. We have added that future research should prioritize validating and adapting dietary assessment tools for use in low-literacy and resource-limited settings to ensure accurate data collection and culturally appropriate nutrition interventions (Lines 374-376).
Comment 8: Include the full search string and exclusion rationale in an appendix.
Response: This is included in the appendix (Supplementary file S2)
Comment 9: Add a simple risk-of-bias matrix or heatmap.
Respond: We appreciate the value of visualizing study quality and understand that a risk-of-bias matrix or heatmap can enhance interpretability in systematic reviews. However, as this is a scoping review, we followed the methodological guidance outlined by the Joanna Briggs Institute (JBI) and the PRISMA-ScR checklist, which recommend that scoping reviews do not routinely include a formal risk-of-bias assessment. This is because the aim of scoping reviews is to map the breadth and nature of evidence rather than to assess the quality or produce synthesized outcomes. Nevertheless, we acknowledge the intent behind the suggestion, and as a compromise, we have included a brief narrative discussion of study design strengths and limitations in the Results or Discussion section to help readers better interpret the evidence base. We hope this addresses your comment while remaining consistent with the accepted scoping review methodology. We were guided by this paper https://journals.lww.com/jbisrir/fulltext/2020/10000/Updated_methodological_guidance_for_the_conduct_of.4.aspx/1000?casa_token=aLTrnWrliXsAAAAA:MGFL4GyZiISQ7W1-XIddWDDCXVxWn-o-8nAS2-SjSs3M9SR33aMqSwZ0Gsv9n2uESpa2bYXGhJ7_Q5mbUpg97qum
Comment 10: Distinguish dietary determinants (e.g., access, knowledge) from dietary patterns (e.g., Western, traditional).
Response: We have revised the Introduction, Results, and Discussion sections to explicitly define and differentiate these concepts in lines 64-69; 212-216; and 304-308. We have also clarified this distinction in the Conceptual Framework (Figure 2), where determinants are positioned as upstream influences shaping dietary behaviors, and patterns are represented as downstream outcomes associated with NCD risk; lines 261-264.
Comment 11: Stratify findings by life stage (e.g., adolescence, pregnancy, postpartum).
Response: results are now stratified by life stages in Table 1; lines 287-288
Comment 12: Use biomarker data to validate self-reported dietary intake.
Response: We acknowledge the critical role of biomarker data in validating self-reported dietary intake, particularly given the limitations of recall-based methods. However, most of the studies included in our review relied on self-reported data, with only a few incorporating biomarker-based assessments. As such, biomarker validation was not a consistent feature across the evidence base and was beyond the scope of our synthesis. We have now highlighted this as a key evidence gap and emphasized the need for biomarker integration in future research within the revised Discussion section; Lines 376-377; 386-388
Comment 12: Recommend a minimum dataset for future cohort studies (e.g., DDS, CRP, HbA1c).
Response: We have added a recommendation in the Discussion section proposing a minimum dataset for future cohort studies, including dietary diversity scores (DDS), biomarkers such as C-reactive protein (CRP) and HbA1c, and anthropometric indicators. These measures will enhance the accuracy and comparability of dietary and health data across diverse settings. Lines 359-362
Comment 13: Address how geographic data gaps affect equity in research and policymaking.
Response: We have expanded the Discussion section to reflect on how geographic data gaps may lead to an overreliance on evidence from a limited number of countries or urban settings. This can result in policy blind spots and interventions that fail to address the diverse dietary patterns, health needs, and structural barriers faced by underserved populations across SSA; lines 365-372.
Comment 14: Propose a visual evidence gap map (e.g., traffic light system by country/theme).
Response: A visual evidence gap map (traffic light system) showing coverage by country in included; lines 284-285.
Comments 15: Methodological gaps relative to JBI 2020 guidance:
- No publicly available protocol (e.g., OSF registration or retrospectively to PROSPERO).
Response: We acknowledge these suggestions and would like to clarify that our scoping review was informed by the Updated JBI methodological guidance for the conduct of scoping reviews, as outlined by Peters et al. (2020), rather than the guidance for systematic reviews or meta-analyses. Scoping reviews serve different purposes than systematic reviews particularly in mapping existing evidence, identifying gaps, and clarifying key concepts without necessarily assessing study quality or synthesizing results quantitatively. We have cited the relevant guidance for transparency: Peters MD, Marnie C, Tricco AC, Pollock D, Munn Z, Alexander L, McInerney P, Godfrey CM, Khalil H. Updated methodological guidance for the conduct of scoping reviews. JBI Evidence Synthesis. 2020;18(10):2119–2126. See lines 89-102
- No justification for selecting a scoping review over a systematic review
Response: A rationale for selecting a scoping review has now been added; lines 83-89
- Lacks documentation of pilot testing for inclusion criteria or data extraction.
Response: We did not conduct formal pilot testing; however, we followed an iterative team-based approach to refine inclusion criteria and charting forms.
- Does not report excluded studies with reasons at full-text screening.
Response: We acknowledge that reporting excluded studies with reasons enhances transparency. In our review, studies were excluded at the full-text screening stage primarily because they did not meet the predefined eligibility criteria, such as population focus, geographical scope, or relevance to dietary determinants and NCD outcomes in women of reproductive age in SSA. While we did not tabulate these exclusions, we followed a rigorous screening process guided by the PCC framework and documented decisions internally.
- No evidence of handsearching or updating the search iteratively.
Response: Our search strategy was limited to electronic databases. We recognize the value of handsearching and iterative updates and will incorporate these in future protocols.
- Unclear whether data extraction involved multiple reviewers or how disagreements were resolved.
Response: This has been clarified. Data extraction was conducted independently by two reviewers, with discrepancies resolved through discussion or third-party consultation; lines 128-131.
- No structured synthesis framework (e.g., mapping by PCC or life stage).
Response: A structured synthesis by life stage has now been included in Table 1; lines 287-288
- Exclusion of non-English studies is unjustified.
Response: We acknowledge the importance of including studies in multiple languages to ensure comprehensive coverage. However, our team is proficient only in English and our respective vernacular languages. As a result, we are unable to accurately interpret and analyze studies published in other languages. To maintain the integrity and accuracy of our review, we have limited our inclusion criteria to English-language studies. We believe this approach ensures that our findings are based on data we can thoroughly understand and evaluate.
- Lacks visual/tabular synthesis of core concepts and contexts.
Response: A visual summary figure and structured table have now been added to synthesize key concepts in Table 1 and figure 3.
- No critical appraisal, even informally, to guide interpretation.
Response: While a formal critical appraisal was not conducted, we have now included a comprehensive narrative overview of the study limitations. This overview addresses key methodological and contextual limitations, such as small sample sizes, short study durations, reliance on self-reported dietary data, and limited geographic scope. Highlighting these limitations, provides a clearer context for interpreting the findings and to guide future research efforts in this area.
Round 2
Reviewer 3 Report
Comments and Suggestions for Authors
I think the authors attempted to address many aspects and I really appreicate the topic of NCDs in women of reproductive age in SSA as it is a critical and under-researched area. I think the socio-ecological framework is a good choice and fits both JBI mapping and systemic thinking and good use of PRISMA-ScR, but I noticed inconsistencies in applying the JBI methodology thoroughly.. also some other issues still persist..
- I saw that most included studies (15 of 19) are cross-sectional, which limits causal interpretation.
- I would have expected greater use of biomarker-based assessments, especially for outcomes like inflammation and micronutrient status.
- I found the term “dietary determinants” used too broadly without operational definitions or typological breakdown.
- I think stratification is missing—by age, urban/rural setting, HIV status, or dietary assessment method.
- I felt the synthesis leaned more toward narrative than structured mapping.
- I would have preferred clearer visual summaries, like evidence matrices or heat maps.
- I found the PRISMA flow diagram hard to interpret… low graphic quality, it’s crowded, poorly labeled and doesn’t clearly track the screening process. I noticed some typos (eg “tow countries”) and number reading issues that affect clarity and professionalism.
- I think the manuscript would benefit from a summary table of study strengths, limitations and methods used.
- I find the policy implications too vague…there’s little linkage to specific frameworks (eg WHO PEN, national dietary guidelines)
- I would expect more specificity and a clearer bridge between evidence quality and implementation feasibility.
- I would have liked more actionable framing of food fortification gaps… what’s missing, where and why it matters.
- I recommend designing cohort studies with standardized biomarkers (eg, CRP, HbA1c, iron indices…) and context-adapted dietary tools.
- I think adolescent women are underrepresented in the analysis and deserve a focused section.
- I suggest outlining a minimum data package for NCD-focused nutrition research in SSA, to guide future efforts more systematically.
Overall, I see this review as a good first attempt to map the evidence. But I think it needs more methodological clarity, stronger data visualization and more targeted policy translation. With tighter definitions, improved formatting and more critical appraisal of included studies, this work could have real regional impact.
Comments on the Quality of English LanguageIn the PRISMA (tow)
Author Response
- I saw that most included studies (15 of 19) are cross-sectional, which limits causal interpretation.
Response: We agree with the reviewer that the predominance of cross-sectional designs (15 out of 19 studies) is a methodological limitation that restricts causal inference. This was acknowledged in the methods, discussion and conclusion sections, where we highlighted the need for future longitudinal cohort studies to better elucidate temporal relationships between dietary exposures and NCD outcomes. In response to the comment, we have further emphasized this limitation in the revised manuscript and included a recommendation for future studies to incorporate longitudinal and biomarker-based methodologies to enhance causal interpretation and policy relevance (lines 261-263; 449-454; 493)
- I would have expected greater use of biomarker-based assessments, especially for outcomes like inflammation and micronutrient status.
Response: We appreciate this valuable observation. The limited use of biomarker-based assessments in the included studies represents a critical gap in the current evidence base. As detailed in Section 3.2, only one study exclusively used biomarkers, while four combined biomarker data with self-reported dietary assessments (lines 259-263). This methodological limitation is discussed in the manuscript, and we recommend that future research integrate objective biomarker data, such as C-reactive protein (CRP), HbA1c, and micronutrient panels, to more accurately assess inflammation and nutrient status (lines 422-425; 454-456). This recommendation is also reflected in the Conclusion section, emphasizing the importance of biomarker-informed approaches to strengthen causal interpretation and policy relevance (lines 493 - 497).
- I found the term “dietary determinants” used too broadly without operational definitions or typological breakdown.
Response: Thank you for this important observation. We have addressed this by incorporating an operational definition of “dietary determinants” in the Introduction (paragraph 3), where we define them as contextual and behavioral factors that influence food choices. We also elaborated on the typologies, including individual-level factors (e.g., affordability, nutrition knowledge), environmental influences (e.g., urbanization, food insecurity), and sociocultural norms (lines 65-69). These distinctions are now more explicitly reflected in the Results (Sections 3.2–3.3) (lines 241-243 and 270-272) and visually supported by the conceptual framework in Figure 2 (lines 311-312)
- I think stratification is missing by age, urban/rural setting, HIV status, or dietary assessment method. I felt the synthesis leaned more toward narrative than structured mapping.
Response: Stratification by age group, geographic setting, HIV status, and dietary assessment method was partially addressed in our results, particularly in Section 3.1, where urban versus rural differences and HIV-specific subgroups were mentioned. In response to this feedback, we have updated Table 1 (lines 338-339) to include these stratification variables. Additionally, we have added a recommendation in the Discussion section for future studies to incorporate stratified analyses to better inform targeted and context-specific interventions (lines 393-397).
- I would have preferred clearer visual summaries, like evidence matrices or heat maps.
Response: While the scoping review methodology was designed to accommodate heterogeneity in study designs and outcomes, necessitating a predominant narrative synthesis, we recognize the value of structured visual mapping. In response, we revised Table 2 title to more clearly align dietary factors with NCD outcomes and analytic methods (lines 342). Additionally, Figures 2 and 3 were updated to more explicitly reflect multilevel determinants and outcome domains (lines 311 and 325). These enhancements aim to improve the clarity and accessibility of the evidence structure while preserving narrative depth
- I found the PRISMA flow diagram hard to interpret… low graphic quality, it’s crowded, poorly labeled and doesn’t clearly track the screening process.
Response: We have revised the PRISMA flow diagram to improve its visual clarity, labeling, and overall quality. The updated figure now more clearly illustrates the identification, screening, eligibility, and inclusion stages of the review process, addressing issues of crowding and interpretability (lines195-199).
- I noticed some typos (eg “tow countries”) and number reading issues that affect clarity and professionalism.
Response: In response, we have corrected all noted typographical errors, including the misspelling of “tow countries,” and addressed formatting issues related to numerical data. These revisions were made to enhance clarity, readability, and overall professionalism throughout the manuscript. The paper was once again revised by an English speak editor.
- I think the manuscript would benefit from a summary table of study strengths, limitations and methods used.
Respond : We appreciate the reviewer’s suggestion. However, we believe that the methods used in the included studies are already clearly outlined in Table 1, which details study design, setting, population characteristics, and dietary assessment methods. Additionally, limitations are discussed in depth in the Discussion section. To enhance balance, we have added a summary of study strengths just before the limitations paragraph in the Discussion section (lines 415-420).
- I find the policy implications too vague…there’s little linkage to specific frameworks (eg WHO PEN, national dietary guidelines)
Respond : While we aimed to provide broad policy recommendations applicable across diverse SSA contexts, we acknowledge the need for clearer links to existing frameworks. We have therefore added a sentence in the Discussion section referencing relevant policy instruments such as the WHO Package of Essential Noncommunicable Disease Interventions (WHO PEN) and national dietary guidelines, to strengthen the applicability and alignment of our recommendations lines 463-466).
- I would expect more specificity and a clearer bridge between evidence quality and implementation feasibility.
Respond : Definitely, linking evidence quality to implementation feasibility is essential. While the Discussion section outlines key limitations and methodological gaps, we have now added a bridging statement to clarify how the quality of evidence, particularly the reliance on self-reported data and limited biomarker use, affects the feasibility and design of nutrition interventions. This addition aims to strengthen the connection between research findings and practical application (lines 449-457).
- I would have liked more actionable framing of food fortification gaps… what’s missing, where and why it matters.
Respond : Notably, clearer articulation of food fortification gaps would strengthen the policy relevance of the manuscript. While the Discussion section highlights the underutilization of fortification programs, we have now added a more actionable framing that identifies specific barriers, such as limited coverage in rural areas, weak regulatory enforcement, and lack of public awareness, and explains why addressing these gaps is critical for reducing micronutrient deficiencies and NCD risk (line 386-391).
- I recommend designing cohort studies with standardized biomarkers (eg, CRP, HbA1c, iron indices…) and context-adapted dietary tools.
Respond : We agree that future cohort studies should incorporate standardized biomarkers and culturally adapted dietary tools to improve data quality and relevance. While we already highlight the need for biomarker-informed research and validated dietary assessments, we have now added a more specific statement in the Discussion section to emphasize the importance of CRP, HbA1c, iron indices, and context-sensitive dietary instruments in future study designs (lines 493-497).
- I think adolescent women are underrepresented in the analysis and deserve a focused section.
Respond : Yes, adolescent women of reproductive age are underrepresented in the included studies. While we noted this limitation in the Discussion section, we have now expanded the text to more clearly emphasize the need for adolescent-focused research and interventions, and to acknowledge the unique vulnerabilities of this subgroup (lines 472-479).
- I suggest outlining a minimum data package for NCD-focused nutrition research in SSA to guide future efforts more systematically.
Respond: Agreeing that outlining a minimum data package would enhance the utility of the review for guiding future research. While we already recommend the use of biomarkers and dietary diversity scores, we have now added a more structured summary in the Discussion section to specify key components of a minimum dataset for NCD-focused nutrition research in SSA (lines 425-433)
Round 3
Reviewer 3 Report
Comments and Suggestions for Authors
All my comments are not only acknowledged but also concretely reflected in the revised manuscript, either through new figures, tables, revised discussion sections or methodological clarifications. The authors have made a sincere and substantive revision!